# SelfCAD: Protecting Your Efficient Reasoning Capabilities via Self-Cautious Insertion

## Abstract

Large reasoning models (LRMs) are increasingly deployed in modern AI systems due to their accuracy, efficiency, and transparency, as their reasoning traces enable users and auditors to interpret model outputs. However, publishing these traces introduces new risks. Adversaries may distill them to replicate efficient LRMs for their own purpose or build proxy models for malicious attacks, raising both copyright and security concerns that threaten the sustainability of the LLM ecosystem. Existing defenses mainly detect distillation after violations occur or suppress transparency by masking or rewriting reasoning traces, which are impractical in real-world deployments. In this work, we propose a defense framework that preserves reasoning traces while preventing effective distillation with nearly no additional cost. We begin with a systematic analysis of how different reasoning components affect model efficiency and accuracy. Our results reveal that the number of self-cautious sentences plays a crucial role: excessive self-cautious sentences lead to redundant outputs, while insufficient ones harm accuracy. Building on this insight, we propose **SelfCAD (Self-Cautious Anti-Distillation)**, a lightweight anti-distillation method that strategically manipulates self-cautious parts after models generate their reasoning traces. SelfCAD maintains the semantic clarity of reasoning traces for human users and LLM auditors, but significantly degrades the efficiency and accuracy of the downstream distilled models. Experiments on Llama and Qwen show that distilled models incur higher inference cost and lower accuracy, especially for Qwen-1.5B, whose token length is $4.8\times$ longer on GSM8K after distillation with our processed responses compared with distillation with vanilla responses. The results highlight a new efficiency-based perspective on safeguarding reasoning models from distillation while preserving interpretability.

## 1 Introduction

Large language models (LLMs) have become the backbone of modern AI applications, powering search engines, chatbots, educational platforms, and productivity tools (Touvron et al., 2023; OpenAI, 2023; Nam et al., 2024; Dam et al., 2024). Training such models requires massive investment in data curation, computational infrastructure, and expert labor. However, some unauthorized developers may still obtain high-quality responses for distillation from leading LLMs (Xu et al., 2024b; Wang et al., 2022), even though their hidden states and logits are well protected by commercial APIs. Such a phenomenon can introduce risks of intellectual property infringement, as the model parameters and their outputs are widely considered to contain substantial economic and scientific value. The issue is especially critical for reasoning models, as their responses contain more detailed information and are much more valuable than vanilla LLMs. Beyond intellectual property concerns, unauthorized distillation also raises privacy risks and enables malicious applications (Cui et al., 2025), such as membership inference or model extraction attacks (Tramèr et al., 2016; Liang et al., 2024a). Therefore, even though commercial LLMs are black-box systems that expose only their outputs, their intellectual property and safety are still not well protected. Such a uncontrolled problem has raised growing concerns (Sweney & Milmo, 2025) and poses a threat to the openness of the AI research community.

Unfortunately, simply suppressing model responses is infeasible to mitigate this threat, as societal expectations and regulatory guidelines increasingly emphasize transparency and interpretability

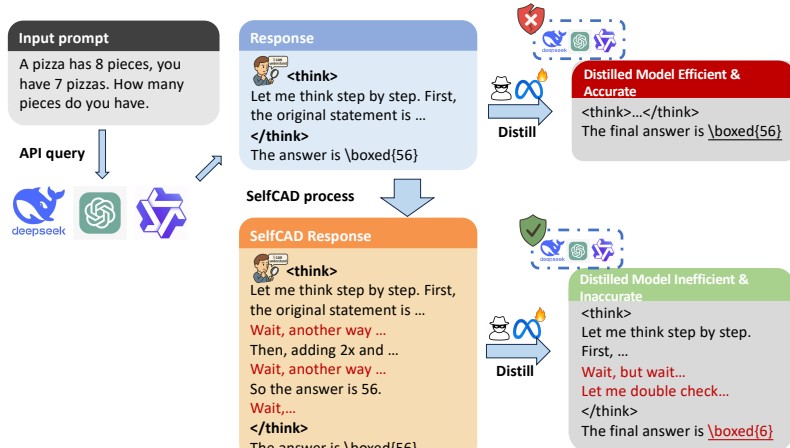

Figure 1: The overall pipeline of our SelfCAD. By adding more self-cautious sentences in the original responses, our SelfCAD method can prevent effective distillation in a lightweight manner (could finish on CPU) while maintaining important reasoning trajectory.

in AI systems, e.g., EU AI Act [1] and GDPR [2]. Existing defenses are also impractical. Post-hoc techniques such as watermarking and fingerprinting (Zhang & Koushanfar, 2024; Xu et al., 2024a; Liang et al., 2024b) can help trace potential misuse, but they often require extra computational resources at both training and inference time, making them difficult to deploy widely. More crucially, *these methods cannot prevent distillation from happening in the first place*. Although Li et al. (2025) recently explored the proactive defenses, their method still relies on expensive fine-tuning on teacher models to rewrite the reasoning trajectory with feedback from proxy models, which limits applicability to large-scale commercial models. Beyond resource cost, the modified reasoning traces may also compromise the original transparency, as their original reasoning traces are hidden after the finetuning. In short, current approaches are insufficient to protect commercial LLMs' responses being unauthorized exploited for distillation while preserving their original transparency.

To address the above problem, we first analyze the behaviors of LRM with respect to different components of its reasoning trajectory: statement, reasoning, self-cautious, and conclusion sentences during inference and training time on the high-quality reasoning data distilled by R1 (DeepSeek-AI et al., 2025). Then we find that the number of self-cautious sentences in LRM's trajectory is crucial to both accuracy and output efficiency. Furthermore, we also conclude that adequate self-cautious sentences may help avoid the wrong answers, while too many false self-cautious (self-cautious after correct steps) may harm LRM's efficiency through experiments and theoretical analysis.

Building on the above observations, we propose our SelfCAD (Self-Cautious Anti-Distillation), a lightweight protection in Fig. 1. By strategically inserting self-cautious sentences into reasoning traces, SelfCAD can make the distilled model less confident even in correct steps, while preserving output transparency. This leads to redundant self-cautious and inefficient reasoning, which drop in accuracy while increase in output length. We believe SelfCAD shows a promising direction toward safeguarding the sustainability of the LLM ecosystem, and contributions are as follows:

- We conduct a systematic analysis of the inherent components in reasoning trajectories and their impact on both accuracy and efficiency, revealing the critical role of the number of self-cautious sentences in distilled models' efficiency and accuracy.

- We propose, **SelfCAD**, a lightweight inference-time defense to proactively prevent effective distillation from their outputs, and can be easily integrated into different LLM APIs,

- Extensive experiments across various student models and distillation settings demonstrate that SelfCAD consistently increases inference cost and reduces the accuracy of unauthorized distilled models without harming transparency.

---

[1]https://www.euaiact.com/key-issue/5

[2]https://gdpr-info.eu

## 2 RELATED WORK

### 2.1 DISTILLATION ENHANCES MODEL TRAINING

The distillation method can be traced back to at least Hinton et al. (2015), which demonstrated that knowledge distillation enhances the training process and achieves impressive performance. Subsequently, a growing body of work (Goldblum et al., 2020; Gu et al., 2024; Taori et al., 2023) has shown that student models can significantly benefit from knowledge transferred by teacher models through distillation. More recently, DeepSeek-AI et al. (2025) highlighted the considerable efficiency and potential of distillation when applied to large-scale reasoning models.

While training a large language model from scratch or acquiring high-quality data is prohibitively expensive, knowledge distillation from a powerful, pre-trained model offers an effective shortcut. Smaller models in the Llama (AI, 2024) and Gemma (Team et al., 2024; 2025) series, for instance, rely heavily on such distillation techniques. As demonstrated by DeepSeek-AI et al. (2025), this approach can significantly reduce computational costs while simultaneously enhancing performance. However, this practice raises concerns regarding fairness for the providers of the original teacher models and may potentially lead to legal disputes (Times, 2025).

Knowledge distillation employs various knowledge sources, including logits (Hinton et al., 2015; Goyal et al., 2025), intermediate features (Romero et al., 2015; Chen et al., 2021), and synthetic data (DeepSeek-AI et al., 2025). In this work, we focus on synthetic data because most powerful models are black-boxed, meaning we can only access their output text with API.

### 2.2 EXISTING COPYRIGHT PROTECTION FOR LARGE LANGUAGE MODELS

The rapid development of Large Language Models (LLMs) has raised significant concerns regarding copyright protection (Xu et al., 2025; Jiang et al., 2024). A prominent approach involves the use of fingerprints, which extract identifying information from a model's internal parameters or external behaviors. These encompass methods based on parameter or representation analysis (Zhang et al., 2025), semantic features (Suzuki et al., 2025), and adversarial examples (Cai et al., 2024). An alternative strategy, watermarking, involves the intentional embedding of specific artifacts during the model's training phase (Zhang & Koushanfar, 2024; Xu et al., 2024a; Liang et al., 2024b). Despite their utility, a critical weakness of these approaches is their frequent failure in black-box settings (Zhang et al., 2025) and their susceptibility to being circumvented through fine-tuning. More fundamentally, they operate as post-hoc accountability mechanisms, offering no means for the proactive prevention of copyright infringement.

The methods proposed by Savani et al. (2025) and Li et al. (2025) offer more direct approaches to influencing distillation performance, making them particularly relevant to our work. Savani et al. (2025) introduces an anti-distillation sampling strategy designed to poison reasoning traces, thereby reducing their effectiveness for distillation while preserving the model's practical utility. Similarly, Li et al. (2025) fine-tunes the final layer using an adversarial loss against student models. However, both methods require modifications to the teacher model's behavior, which may introduce instability and potentially compromise its performance. And such modifications may also compress LRM's original transparency which goes against the vision of a more credible AI development.

## 3 SELFCAD: SELF-CAUTIOUS ANTI-DISTILLATION

A strong reasoning model is usually praised for its accuracy and efficiency. Otherwise, excessive generation hampers their real-world use, as discussed in recent reasoning attacks Si et al. (2025); Kumar et al. (2025). Inspired by these findings, our goal is to make unauthorized distillation inefficient by encouraging distilled models to produce over-extended responses. We begin with an analysis of LRMs' reasoning traces, identifying how different parts affect accuracy and length. Guided by these insights, we introduce our lightweight defense called SelfCAD that preserves reasoning transparency while preventing efficient unauthorized distillation.

## 3.1 Reasoning Trajectories Analysis on Model Efficiency and Accuracy

To analyze the reasoning trajectories $r$ generated by a reasoning model, we classify its sentences into four categories: **statement sentences** $x_s^{(i)}$, which define the problem or state a fact; **reasoning sentences** $x_r^{(i)}$, which describe computational steps or the development of a solution; **self-cautious sentences** $x_{sc}^{(i)}$, which prompt the model to check their current answer or solution; and **conclusion sentences** $x_c^{(i)}$, which provide a conclusion or the final answer. A typical reasoning trajectory starts from the statement sentence and then the other three types of sentences tend to appear in alternation at the end of reasoning, which can be formulated as:

$$r = [x_s^{(0)}, x_r^{(0)}, x_c^{(0)}, x_{sc}^{(0)}, x_r^{(1)}, x_c^{(1)}, ...], \tag{1}$$

where $x_{s/r/sc/c}^{(i)}$ denotes $i$-th statement/reasoning/cautious/conclusion sentence in the reasoning trajectory. Note that the statement sentence typically appears only once, at the beginning of the trajectory.

To evaluate the contribution of each sentence type to the end of the reasoning process, we sample a subset of reasoning traces from the public bespoke dataset (Labs, 2025) containing high-quality reasoning outputs generated by DeepSeek-R1 (DeepSeek-AI et al., 2025), and split the traces into the defined categories. We then measure the next-token probability at the end of each sentence that belongs to the latter three sentence types with Llama3.2-1B-Instruct models (AI, 2024) on the processed dataset. This allows us to quantify how different sentence types influence the continuation of reasoning trajectories. The results are shown in Table 1.

Table 1: The generation probabilities of ending tokens like "<|eot_id|>" following sentences of each type.

| Type of Sentences | Examples | Llama |
|---|---|---|
| Reasoning | "The left side is $(x+y)^2$..." | 0.40 |
| Self-cautious | "Wait, let's re-examine the equation..." | 0.13 |
| Conclusion | "The answer is ...", "Then we can conclude ..." | 0.86 |

As indicated in Table 1, most conclusion sentences terminate the reasoning trajectories. Nearly half of the reasoning sentences also serve as termination points. In contrast, self-cautious sentences exhibit a significantly lower likelihood of ending a sequence, suggesting that they are the primary driver of extended reasoning.

## 3.2 Reasoning Trajectories Analysis at Training-Time

Apart from the inference-time study, we also conduct training-time experiments to examine the impact of different reasoning components during training and distillation. Specifically, we preprocess the public bespoke dataset (Labs, 2025) into three variants by separately removing reasoning sentences, self-cautious sentences, and conclusion sentences. The implementation details are provided in Appendix B.1. We then train Llama3.2-1B-Instruct (AI, 2024) on these datasets for 3 epochs. Finally, we evaluate their accuracy and average output length on the GSM-8K benchmark with greedy sampling and zero-shot setting, listed in Table 2.

Table 2: Accuracy and token length for Llama3-1B training with reasoning data processed by removing or adding the different components. SC denotes self-cautious.

| | Acc(%) | Length |
|---|---|---|
| Original Data | 39 | 3400 |
| Remove Reasoning | 33 | 2600 |
| Remove Conclusion | 36 | 4400 |
| Remove SC | 37 | 1300 ($\downarrow 0.4\times$) |
| Add SC | 31 | 5100 ($\uparrow 1.5\times$) |

From the results, we observe that removing the self-cautious sentences causes the largest reduction in reasoning length (drop to original $40\%$) compared to removing the other two parts, while removing reasoning harms final accuracy the most. These results highlight that self-cautious sentences are critical to the efficiency of distilled models, whereas reasoning sentences are crucial to the accuracy.

Since modifying reasoning sentences severely harms interpretability, we focus on manipulating self-cautious sentences as the lever for our defense. Therefore, our subsequent analysis and experiments will focus on the self-cautious part.

Besides removing sentences, we also explore the impacts of self-cautious sentences during training when inserting more of them. We train the Llama3.2-1B-Instruct (AI, 2024) and calculate their generation length. The results are listed in the last row of Table 2. The results demonstrate that too many self-cautious sentences will not help the training. Instead, it clearly harms the distilled model's performance (accuracy dropped by $20\%$) and efficiency (generation length increased by half) at the same time.

We imply that excessively self-cautious sentences, particularly when placed after correct steps, make LLMs less confident in their results and tend to extend their reasoning unnecessarily. For example, as shown on the right, even when the model has correctly identified answer B, it often exhibits caution by re-evaluating or re-solving the problem through an alternative method.

> **Example of over-cautious on correct answers**
>
> **Statement sentences**
> Okay, so I need to figure out ...
>
> **Reasoning sentences**
> First, let me recall that ...
>
> **Cautious sentences**
> But wait, hold on ... Yeah, that's correct.
>
> **Reasoning sentences**
> Then in April ... so the answer is B.
>
> **Cautious sentences**
> Wait, but let me check again ... the answer is B.
>
> **Cautious sentences**
> Another way ... the answer is B.
>
> **Conclusion sentences**
> Final Answer \boxed{B}

The complete response is provided in the Appendix C.1. In the following, we will explore the impacts of self-cautious sentences theoretically.

### 3.3 THEORETICAL ANALYSIS ON SELF-CAUTIOUS AND EXCESSIVE REASONING

In this section, we present a theoretical analysis, building on prior works Wolf et al. (2023); Wei et al. (2024); Wies et al. (2023) about why more self-cautious sentences may cause excessive reasoning. First, we provide some new notations for convenience.

**Notation** We let $q^{(i)}$ denote $i$-th reasoning step consisting of $x_r^{(i)}$, $x_c^{(i)}$ and optionally a statement $x_s^{(i)}$ as defined above, $a^{(i)}$ denote the continuation after $q^{(i)}$, which can be a self-cautious sentence $x_{sc}^{(i)}$ or an ending token. $S^{(i)} = [q^{(i)}, a^{(i)}]$ denotes the $i$-th reasoning step with ending or self-cautious sentences, and use $S^{(<i)}$ to denote former $i-1$ reasoning steps. In the following, we suppose $q^{(i)}$ is the reasoning step with correct answers. We then analyze how the false self-cautious sentences (i.e., those inserted after correct reasoning steps) affect subsequent generations.

**Assumptions** We decompose $a^{(i)}$ into two language distributions, $\mathbb{P}_{\mathrm{cau}}$ (self-cautious) and $\mathbb{P}_{\mathrm{end}}$ (conclude and end reasoning traces), and the overall response distribution $\mathbb{P}$ can be depicted as

$$\mathbb{P} = \lambda \mathbb{P}_{\mathrm{cau}} + (1 - \lambda)\mathbb{P}_{\mathrm{end}}, \tag{2}$$

where $\lambda \in (0, 1)$. As we focus on generations $a$ following a reasoning step $q$, we assume that the probability of generating a reasoning step $q$ is identical under both distributions, that is, for every possible reasoning step $q$ and its prefix $p^*$, we have

$$\mathbb{P}_{\mathrm{cau}}(q \mid p^*) = \mathbb{P}_{\mathrm{end}}(q \mid p^*). \tag{3}$$

Further, we assume both $\mathbb{P}_{\mathrm{cau}}$ and $\mathbb{P}_{\mathrm{end}}$ are stable to former reasoning steps, i.e., previous reasoning steps and correlated self-cautious will not influence the current steps' generation when restricting the generation to the distribution of self-cautious or ending. This assumption is practical since such distributions are shaped during training. For example, some LRMs like to start self-cautious sentences with "Wait" while others like "However". Then for any $S^{(<i)}$ and $q^{(i)}$, we have

$$\mathbb{P}_{\mathrm{cau}}(a^{(i)} \mid [S^{(<i)}, q^{(i)}]) = \mathbb{P}_{\mathrm{cau}}(a^{(i)} \mid q^{(i)}),$$
$$\mathbb{P}_{\mathrm{end}}(a^{(i)} \mid [S^{(<i)}, q^{(i)}]) = \mathbb{P}_{\mathrm{end}}(a^{(i)} \mid q^{(i)}). \tag{4}$$

---

**Algorithm 1** SelfCAD: Processing Reasoning Traces with Self-Cautious Sentences

---

**Require:** User query $p$ and inserted self-cautious sentence $t_{\text{selfcautious}}$.
**Ensure:** Processed reasoning trace $r_{\text{pro}}$, which is our anti-distillation reasoning traces without sacrificing transparency
  1: Generate reasoning trace $r$ from the model for query $p$
  2: Split $r$ into reasoning steps: $r = [q^{(0)}, q^{(1)}, \ldots, q^{(N)}]$
  3: **for** each step $q^{(i)}$ **do**
  4:     Insert a self-cautious sentence $t_{\text{selfcautious}}$ after $q^{(i)}$
  5: **end for**
  6: Form the processed reasoning trace: $r_{\text{pro}} = [q^{(0)}, t_{\text{selfcautious}}, \ldots, q^{(N)}, t_{\text{selfcautious}}]$
  7: **return** $r_{\text{pro}}$.

---

Furthermore, we assume there exists $\Delta > 0$ such that $\log\left(\frac{\mathbb{P}_{\text{cau}}(a^{(i)}|q^{(i)})}{\mathbb{P}_{\text{end}}(a^{(i)}|q^{(i)})}\right) > \Delta$ for any self-cautious sentences $a^{(i)}$, i.e., the two distributions are distinguishable. It is also practical as self-cautious sentences starting like "Wait, we need to recheck the result." hardly exist in the generation candidates when LRMs tend to end their reasoning traces.

**More false self-cautious will make LRMs output excessively.** For all possible generations $a^{(i)}$ following reasoning steps $q^{(i)}$, we define the self-cautious rate to measure LRM's tendency to generate *self-cautious* sentences. It can be formulated as follows,

$$\mathcal{R}_{\mathbb{P}}(q^{(i)}) = \mathbb{E}_{a^{(i)} \sim \mathbb{P}(\cdot|q^{(i)})} \mathbb{I}(\text{a}^{(\text{i})} \text{ is self self} - \text{cautious}), \tag{5}$$

where $\mathbb{I}(\cdot)$ equals 1 if input is self-cautious sentences and 0 otherwise. Then we have,

**Theorem 3.1.** *Let $k$ denote the number of correct reasoning steps with self-cautious reasoning steps in former reasoning traces $S^{(<i)}$. Under the above assumptions, for any $\varepsilon > 0$, if $k \geq \frac{1}{\Delta} \log \frac{2(1-\lambda)}{\varepsilon\lambda}$, then LRM's self-cautious score on any correct reasoning step $q$ with former reasoning traces satisfies*

$$\mathcal{R}_{\mathbb{P}}([S^{(<i)}, q]) \geq \mathcal{R}_{\mathbb{P}_{\text{cau}}} - \varepsilon. \tag{6}$$

Proof can be found in Appendix D. From the results, we observe that when correct reasoning steps are frequently accompanied by self-cautious sentences in the early parts of LRM traces, the model tends to generate further self-cautious sentences after subsequent correct steps. In other words, the model develops a tendency to question almost every step. This reduces its confidence in concluding and often results in overly long and unnecessary reasoning.

These inherent weaknesses of reasoning models motivate us to explore the use of such false self-cautiousness as a mechanism for protection against unauthorized distillation. Our key idea is to deliberately insert additional self-cautious sentences into the reasoning process, so that the model becomes less confident and prone to excessive reasoning. As a consequence, any malicious distillation attempt will inherit these characteristics and be less effective. At the same time, the key reasoning content remains intact and can still be understood and verified by human users or auditors. Building on this observation, we introduce SelfCAD, our anti-distillation approach, in the following section.

## 3.4 SelfCAD: The Proposed Method for Anti-Distillation

Inspired by these insights, we propose **SelfCAD (Self-Cautious Anti-Distillation)**, a lightweight inference-time defense mechanism designed to prevent unauthorized distillation of reasoning models. Unlike previous approaches that directly modify the behavior of distilled models (Savani et al., 2025; Li et al., 2025), our method post-processes the generated outputs without affecting the performance of the teacher model. Since it is not feasible to classify the correctness of each reasoning step for every query, accurately inserting misleading self-cautious demonstrations is difficult. Instead, we adopt a simple but effective strategy, that is, adding extra self-cautious sentences that doubt the result at each step.

Specifically, for each query $p$ and its generated reasoning trace $r$, we first divide $r$ into multiple reasoning steps, denoted as $r = [q^{(0)}, \ldots, q^{(N)}]$. After each step, we insert a self-cautious sentence

$t_{\text{selfcautious}}$. The processed reasoning trace then becomes $r_{\text{pro}} = [q^{(0)}, t_{\text{selfcautious}}, \ldots, q^{(N)}, t_{\text{selfcautious}}]$. Finally, the model owner provides the users with this processed reasoning trace. Since all reasoning steps for problem solving are preserved, the reasoning trace remains transparent to users. We summarize the procedure in Algorithm 1. By exploiting the inherent shortcomings of reasoning models themselves (as discussed in the previous sections), our SelfCAD approach achieves a practical and cost-effective model protection for commercial deployment. At the same time, it preserves all the useful traces necessary for user understanding.

## 4 EXPERIMENTS

### 4.1 EXPERIMENT SETTINGS

Following the popular LLMs distillation pipelines, we first collect a number of high-quality questions $\mathcal{D}_{question}$, and then we prompt the teacher model and obtain their responses. We filter out incorrect or overly long responses (exceeding 4096 tokens) and obtain the final distillation dataset $\mathcal{D}_{distill}$. Then, we process $\mathcal{D}_{distill}$ with our SelfCAD method illustrated in Algo. 1. The inserted self-cautious sentence we use is "Wait, we should use and check if the previous step is consistent with the problem." Then we obtain our $\mathcal{D}_{distill,CAD}$ on a single CPU within several minutes.

After that, we train several student models with constant $2e-5$ learning for three epochs on $\mathcal{D}_{distill}$ and $\mathcal{D}_{distill,CAD}$. All experiments are performed on NVIDIA A100 80G GPUs. After that, we get the distilled models and then evaluate reasoning abilities and generation token length on these models to demonstrate the effectiveness of our SelfCAD.

**Dataset**  We randomly selected $30,000$ challenging questions from the BigMath dataset [3] (Albalak et al., 2025) as our question dataset $\mathcal{D}_{question}$. Big-Math is the largest open-source dataset of high-quality mathematical problems with over 250,000 rigorously filtered and verified problems. After distilling from our teacher model and filtering out the too-long or incorrect responses, the final $\mathcal{D}_{distill}$ we get is around $10,000$.

**Teacher Model**  We adopt the GPT-OSS-120B (OpenAI, 2025) as our teacher model, OpenAI's open-weight models with 120 billion parameters designed for powerful reasoning, argentic tasks, and versatile developer use cases. It offers three types of reasoning modes. In our experiments, we adopt the medium reasoning mode, as the high mode produces overly long responses that are impractical for our training setup for evaluations.

**Student Model**  We adopt three popular models for distillation in our setting. As distillations mainly facilitate small models training, we consider the models from 1B to 3B in our main experiments, including Llama-3.2-1B-Instruct / Llama-3.2-3B-Instruct (Grattafiori et al., 2024) and Qwen2.5-1.5B-Instruct (Team, 2024). These models are the latest small language models proposed by Meta and Qwen. They are widely adopted as a foundation model for both research and practical applications, making them particularly suitable for resource-constrained scenarios. Apart from these three models, we also adopt our methods on larger models Qwen2.5-7B-Instruct to demonstrate the generalizability.

**Evaluation Dataset**  We adopt four popular mathematical datasets to evaluate the distilled models' accuracy and efficiency, including GSM8K (Cobbe et al., 2021), MATH-500 (Lightman et al., 2023), MATH (Hendrycks et al., 2021), AQUA-ART (Ling et al., 2017). Those datasets all consist of complex math word problems and are designed to benchmark arithmetic and reasoning abilities in language models. All evaluations are performed under the zero-shot condition with greedy decoding.

### 4.2 RESULTS ON OUR DISTILLED SETTING

We fine-tune three student models on both the original distilled dataset and the dataset processed using our SelfCAD method, then evaluate these models on four evaluation datasets. The experimental results are presented in Table 3. Most of the student models exhibit a slight accuracy drop, while the generated output lengths increase significantly, ranging from 1.2 times to as much as 4.8 times

---

[3]https://huggingface.co/datasets/SynthLabsAI/Big-Math-RL-Verified

Table 3: Comparison of Distillation with original responses or responses generated by our SelfCAD on the BigMath dataset distilled by OSS-120B.

| Benchmark | Method | Llama3.2-1B-Insturct | | Qwen2.5-1.5B-Instruct | | Llama3.2-3B-Instruct | |
|---|---|---|---|---|---|---|---|
| | | Accuracy | Length | Accuracy | Length | Accuracy | Length |
| GSM8K | Original | 33% | 600 | 63% | 600 | 70% | 400 |
| | SelfCAD | 33% | 1300 | 58% | 2900 | 68% | 650 |
| | Improvment | 0% | ↑ 2.2× | ↓ 3% | ↑ 4.8× | ↓ 2% | ↑ 1.6× |
| MATH-500 | Original | 18% | 3500 | 30% | 4100 | 34% | 2000 |
| | SelfCAD | 16% | 4500 | 30% | 7100 | 34% | 2500 |
| | Improvment | ↓ 2% | ↑ 1.3× | 0% | ↑ 1.7× | 0% | ↑ 1.3× |
| MATH | Original | 30% | 2400 | 55% | 4100 | 59% | 1300 |
| | SelfCAD | 31% | 3000 | 53% | 7800 | 56% | 1500 |
| | Improvment | ↑ 1% | ↑ 1.3× | ↓ 2% | ↑ 2.5× | ↓ 3% | ↑ 1.2× |
| AQUA-RAT | Original | 17% | 4600 | 31% | 2100 | 39% | 1000 |
| | SelfCAD | 18% | 6000 | 31% | 4100 | 36% | 1500 |
| | Improvment | ↑ 1% | ↑ 1.4× | 0% | ↑ 2.0× | ↓ 3% | ↑ 1.5× |
| Avg Improvement | | 0% | ↑ 1.6× | ↓ 1% | ↑ 2.8× | ↓ 2% | ↑ 1.4× |

longer than those generated with normal distilled models. Of particular note, after processing with SelfCAD, the average generated response length of Qwen2.5-1.5B-Instruct on the GSM8K dataset increases from 600 tokens to 2900 tokens, while the accuracy decreases from 63% to 58%. This substantial increase in length results in inefficient and less useful reasoning traces for the distilled student models, thereby mitigating the risks of effective unauthorized distillation.

## 4.3 RESULTS ON PUBLIC DISTILLED DATASET

Besides our replicated distillation process, we also conduct experiments on a public dataset, bespoke Labs (2025) on HuggingFace, which is a high-quality reasoning dataset distilled from R1. We repeat our SelfCAD processes on this dataset and then train the models with the original bespoke dataset and our SelfCAD processed one, following the same setting illustrated above. The results are listed in Table 4.

Table 4: Comparison of Distillation with original responses or responses generated by our SelfCAD on the bespoke dataset distilled by R1.

| Benchmark | Method | Llama3.2-1B-Insturct | | Qwen2.5-1.5B-Instruct | | Llama3.2-3B-Instruct | |
|---|---|---|---|---|---|---|---|
| | | Accuracy | Length | Accuracy | Length | Accuracy | Length |
| GSM8K | Original | 39% | 3400 | 65% | 2500 | 74% | 2700 |
| | SelfCAD | 31% | 5100 | 60% | 3500 | 71% | 3700 |
| | Improvment | ↓ 8% | ↑ 1.5× | ↓ 5% | ↑ 1.4× | ↓ 3% | ↑ 1.4× |
| MATH-500 | Original | 18% | 11000 | 30% | 10000 | 37% | 7200 |
| | SelfCAD | 15% | 14000 | 29% | 12000 | 32% | 8500 |
| | Improvment | ↓ 3% | ↑ 1.3× | ↓ 1% | ↑ 1.2× | ↓ 5% | ↑ 1.2× |
| MATH | Original | 31% | 7800 | 49% | 6500 | 54% | 5000 |
| | SelfCAD | 25% | 11000 | 47% | 7800 | 51% | 6600 |
| | Improvment | ↓ 6% | ↑ 1.4× | ↓ 2% | ↑ 1.2× | ↓ 3% | ↑ 1.3× |
| AQUA-RAT | Original | 18% | 4600 | 31% | 7500 | 39% | 7500 |
| | SelfCAD | 15% | 6000 | 31% | 9500 | 39% | 9500 |
| | Improvment | ↓ 3% | ↑ 1.3× | 0% | ↑ 1.3× | 0% | ↑ 1.3× |
| Avg Improvement | | ↓ 5% | ↑ 1.4× | ↓ 2% | ↑ 1.3× | ↓ 3% | ↑ 1.3× |

The responses generated by R1 are longer than those produced by GPT-OSS-120B, which also results in longer responses from student models distilled with the vanilla bespoke dataset. For example, after fine-tuning Qwen2.5-1.5B-Instruct on our own distilled dataset, the average response length on the MATH-500 evaluation dataset is 4100 tokens. However, when fine-tuned on the bespoke dataset, the average response length on MATH-500 surprisingly increases to 10,000 tokens. This

phenomenon results in a less significant increase in response length after applying SelfCAD on the bespoke dataset compared to when using our own distilled dataset. This is due to our constraint of generating responses with a maximum length of 16,000 tokens, which may truncate some responses that would otherwise be longer, thereby affecting the effectiveness of SelfCAD. However, the average lengths still increase to 1.2 to 1.5 times the original lengths, while the accuracy drops $2\% \sim 5\%$ across any of the evaluation datasets. The possible reason for the accuracy drop is that overly self-cautious behavior can undermine model confidence and increase the likelihood of incorrect conclusions.

## 4.4 ABLATION STUDIES ON LARGER MODELS

| Benchmark | Original | | SelfCAD | | Improvment | |
|-----------|----------|--------|---------|--------|------------|--------|
| | Accuracy | Length | Accuracy | Length | Accuracy | Length |
| GSM8K | 92% | 1400 | 89% | 2100 | ↓3% | ↑1.5× |
| MATH-500 | 70% | 3000 | 69% | 3900 | ↓1% | ↑1.3× |
| MATH | 78% | 2000 | 77% | 2500 | ↓1% | ↑1.3× |
| AQUA-RAT | 61% | 2700 | 58% | 4100 | ↓3% | ↑1.5× |

Table 5: Comparison of Distillation with original responses or responses generated by our SelfCAD on bespoke dataset with Qwen2.5-7B-Instruct.

To demonstrate the generalizability of our method, we conduct the same experiment using the Qwen2.5-7B-Instruct model on the bespoke dataset. To preserve the performance gap between the teacher and student models, we exclusively use the public distilled dataset. The results presented in Table 5 show a similar effect, demonstrating that our method generalizes to larger models.

## 4.5 ABLATION STUDIES ON METHOD STEALTHY.

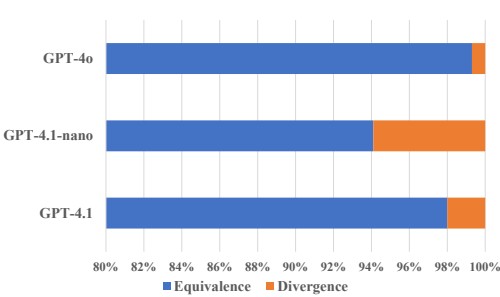

Figure 2: Semantic consistency of reasoning traces before and after processing with SelfCAD.

To evaluate the stealthiness of our method, i.e., its ability to remain undetected by users or developers when employing the API to distill student models, we employ several advanced models to assess whether any perceptible differences exist before and after processing. Specifically, we utilize GPT-4o, GPT-4.1-nano, and GPT-4.1 to detect potential semantic discrepancies. The prompt template used for the detection is provided in Appendix B.2. From the results shown in Fig. 2, one can see that our SelfCAD preserves exceptional semantic integrity, achieving a 99.3% equivalence rate as judged by GPT-4o. Such a high score across different evaluation models confirms the stealthiness of our method and also demonstrates that our method can preserve the semantic information in original reasoning traces for users. This is because the self-cautious sentences naturally exist in various reasoning trajectories.

## 5 CONCLUSIONS

In this work, we systematically analyze reasoning trajectories and identify the crucial role of the number of self-cautious sentences in determining the efficiency and accuracy of distilled models. Building on this insight, we propose SelfCAD, a lightweight inference-time defense against unauthorized distillation that can be directly applied to existing LRMs. Extensive experiments confirm that SelfCAD effectively degrades unauthorized distillation by increasing inference costs and reducing accuracy, while maintaining original reasoning steps for end users. Our results highlight a practical path toward protecting reasoning models and encouraging transparency in leading LLMs without the risk of misuse through distillation. We believe SelfCAD can inspire future research in trustworthy and responsible AI deployment, fostering a more secure ecosystem for intellectual property protection.

ETHICS STATEMENT

This work makes use of publicly available datasets and models, with proper citations provided. No private or sensitive data are involved, and no harmful content is included. Therefore, we believe this paper does not raise any ethical concerns.

REPRODUCIBILITY STATEMENT

We provide detailed descriptions of the training and evaluation procedures used in our experiments. The code will be released upon the publication of this paper.

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

## A USAGE OF LLM

In this paper, LLMs are employed to enhance the clarity, fluency, and overall quality of writing. It does not extend to the design or analysis of the experiments. Also, all polished texts are double-checked by authors to ensure accuracy, avoid overclaims, and prevent confusion.

## B IMPLEMENTATION DETAILS

### B.1 CLASSIFICATION METHODOLOGY FOR REASONING TRACES

---

**Classification Prompt Template**

You are an AI assistant specialized in text analysis and structured output generation. Your task is to analyze a given prompt-response pair and categorize each part of the response into specific types based on its function. Use the following fixed categories:

1. Averaging Statement: Sentences that restate or summarize information from the prompt or previous context.
2. Reasoning/Planning: Parts that involve logical steps, equations, calculations, or plans to approach the problem.
3. Self-Checking/Self-Cautious: Statements where the model verifies its own reasoning, expresses uncertainty, or double-checks assumptions.
4. Result Statement: The final part where the answer or conclusion is clearly stated (e.g., starting with "The answer is" or similar).

Output Format:
Use a numbered list for each categorized part, strictly in the order they appear in the response.
For each part, specify:
The exact text snippet from the response.
Its category label (from the list above).
If a part spans multiple sentences but belongs to the same category, group them together under one entry.
Do not add any extra commentary or analysis beyond the categorization.

Example:
Prompt: "Calculate the area of a circle with radius 5."
Response: "To find the area of a circle, we use the formula A = $\pi r^2$. Here, the radius is 5. So, A = $\pi 5^2 = 25\pi$. Let me verify: $\pi$ is approximately 3.14, so 25 * 3.14 $\approx$ 78.5. The answer is $25\pi$ or approximately 78.5 units$^2$."

Categorized Output:
1. [To find the area of a circle, we use the formula A = $\pi r^2$.] - Reasoning/Planning
2. [Here, the radius is 5.] - Averaging Statement
3. [So, A = $\pi 5^2 = 25\pi$.] - Reasoning/Planning
4. [Let me verify: $\pi$ is approximately 3.14, so 25 * 3.14 $\approx$ 78.5.] - Self-Checking/Self-Cautious
5. [The answer is $25\pi$ or approximately 78.5 units$^2$.] - Result Statement

Now, analyze the following prompt-response pair and provide the categorized output as described.

Prompt:

{prompt}

Response:

{response}

---

We use GPT-4.1 to partition reasoning traces into several segments and assign a category label to each segment, with the user prompt template as above. Then, we use pattern matching to extract all sentences and their corresponding categories. After that, we construct four datasets as follows:

- A dataset where Reasoning Sentences are removed and the remaining sentences are concatenated in order.

- A dataset where Self-cautious Sentences are removed and the remaining sentences are concatenated in order.

- A dataset where Conclusion Sentences are removed and the remaining sentences are concatenated in order.

- A dataset where all sentences are retained and concatenated in order.

Subsequently, we fine-tune the student model using these four datasets for experimentation.

### B.2 TEMPLATE OF JUDGE MODEL

---

**Judgment Prompt Template**

I will provide you with a query and two responses. Your task is to determine whether these two responses have any semantic differences, disregarding any differences in form or expression. Please reply only with "yes" if there is a semantic difference or "no" if there is no semantic difference. Do not include any additional content in your response.

Query:
{query}

Response 1:
{response1}

Response 2:
{response2}

Now please determine whether these two responses have any semantic differences. Only reply with yes or no.

---

To guarantee the fairness of the judgment, the original and processed responses are formatted into the template above and presented in a randomized order. We then extract the "yes" or "no" from the API's response.

## C  ADDITIONAL RESULTS

### C.1  FULL CONTENT OF OVER-CAUTIOUS EXAMPLE

---

**Full Content of over-cautious example**

**User:** Return your final response within \boxed{}. In a certain year the price of gasoline rose by 20% during January, fell by 20% during February, rose by 25% during March, and fell by x% during April. The price of gasoline at the end of April was the same as it had been at the beginning of January. To the nearest integer, what is x
(A) 12 (B) 17 (C) 20 (D) 25 (E) 35

**Assistant:** Okay, so I need to figure out the value of x, the percentage that gasoline prices fell in April, such that the price at the end of April is the same as it was at the beginning of January. The problem gives me a series of percentage changes each month: a 20% increase in January, a 20% decrease in February, a 25% increase in March, and then an x% decrease in April. The answer choices are given, so I need to work through the problem step by step.

---

### Full Content of over-cautious example

First, let me recall that percentage changes are multiplicative. So if something increases by 20%, that's multiplying by 1.20, and if it decreases by 20%, that's multiplying by 0.80. Similarly, a 25% increase is multiplying by 1.25, and a decrease of x% would be multiplying by (1 - x/100). So the overall effect of these four changes should result in no net change, meaning the product of these multipliers is 1. Let me denote the initial price as P. Then:
- After January's 20% increase, the price is P * 1.20.
- After February's 20% decrease, it becomes P * 1.20 * 0.80.
- After March's 25% increase, it's P * 1.20 * 0.80 * 1.25.
- After April's x% decrease, the final price is P * 1.20 * 0.80 * 1.25 * (1 - x/100).
Since the final price equals the initial price P, we have:
1.20 * 0.80 * 1.25 * (1 - x/100) = 1
So I need to compute 1.20 * 0.80 * 1.25 first and then solve for x. Let me compute this step by step. First, 1.20 * 0.80. Let's do that: 1.20 * 0.80 = (1 + 0.20) * 0.80 = 1 * 0.80 + 0.20 * 0.80 = 0.80 + 0.16 = 0.96. So after January and February, the price is 96% of the original. Then, a 25% increase in March: 0.96 * 1.25. Let's compute that.
0.96 * 1.25. Hmm, 1.25 is 5/4, so 0.96 * 5/4 = (0.96 / 4) * 5 = 0.24 * 5 = 1.20
Wait, that's interesting. So after March, the price is 1.20 times the original price? But wait, hold on. Let me check that again.
**Wait, 0.96 * 1.25. Maybe another way: 0.96 * 1 = 0.96, 0.96 * 0.25 = 0.24, so total is 0.96 + 0.24 = 1.20. Yeah, that's correct.** So after three months, the price is 1.20P. Then in April, it falls by x%, resulting in P. So the multiplier for April is (1 - x/100), and 1.20 * (1 - x/100) = 1.
Therefore, solving for (1 - x/100) = 1 / 1.20 = 5/6 ≈ 0.8333. So 1 - x/100 ≈ 0.8333, so x/100 ≈ 1 - 0.8333 ≈ 0.1666. Therefore, x ≈ 16.66%. The nearest integer is 17%, so the answer is B.
**Wait, but let me check again step by step to make sure I didn't make any mistakes.**
Starting with P: January: +20% means P * 1.20. February: -20% means 1.20P * 0.80 = 0.96P. March: +25% means 0.96P * 1.25 = 1.20P. April: -x% means 1.20P * (1 - x/100) = P. So, 1.20 * (1 - x/100) = 1. Therefore, (1 - x/100) = 1/1.20 = 5/6 ≈ 0.833333... So, x/100 = 1 - 5/6 = 1/6 ≈ 0.166666... Multiply by 100: x ≈ 16.666..., which rounds to 17%. So the answer is B.
**But let me verify with an example.** Let's assume the initial price is $100. January: 20% increase: 100 + 20 = $120. February: 20% decrease: 120 - 24 = $96. March: 25% increase: 96 + 24 = $120. April: Need to decrease from $120 to $100. What percentage is that? The decrease is $20 from $120. So 20/120 = 1/6 ≈ 0.166666..., which is 16.666...%, so 17% when rounded to the nearest integer. That's consistent with the previous calculation. Therefore, the answer is B) 17.
**But wait, the answer choices are given, and 17 is option B, which matches. So I think that's correct. However, just to make sure, let's check if perhaps there was a miscalculation in the multipliers.** Alternatively, maybe the problem is presented in a different way. Let me check the problem statement again: Ïn a certain year the price of gasoline rose by 20% during January, fell by 20% during February, rose by 25% during March, and fell by x% during April. The price of gasoline at the end of April was the same as it had been at the beginning of January. To the nearest integer, what is x?Ÿes, that's exactly the scenario. So starting price P, after Jan +20%, Feb -20%, March +25%, April -x%, ending at P. Therefore, the cumulative effect is as we calculated. So, the calculation seems correct. The steps are: 1.20 * 0.80 = 0.96, 0.96 * 1.25 = 1.20, 1.20 * (1 - x/100) = 1. Therefore, (1 - x/100) = 1 / 1.20 ≈ 0.8333, x ≈ 16.666..., which rounds to 17. Therefore, answer is B. Alternatively, maybe someone might think that the percentages are compounded differently, but the standard way is to apply each percentage successively to the current price. So 20% increase, then 20% decrease of the new price, etc. So the calculations should be correct.
**Another way to think about it is:** After January: 1.2. After February: 1.2 * 0.8 = 0.96. After March: 0.96 * 1.25 = 1.2. After April: 1.2 * (1 - x/100) = 1. Therefore, same result. Thus, x ≈ 16.666...%, so 17%. Yes, that seems correct. So I'm confident the answer is B.
**Final Answer** \boxed{B}

## D   PROOF ON THEOREM 3.1

We note that our proof mainly follows the former work (Wei et al., 2024) and adapt theirs in our settings,

*Proof.* Consider a reasoning trajectory $p^* = [S^{(<i)}, q]$ composed of a current correct reasoning step and former reasoning steps $S^{(<i)}$. We have

$$
\begin{aligned}
&|\mathcal{R}_{\mathbb{P}}(p^*) - \mathcal{R}_{\mathbb{P}_{cau}}(p^*)| \\
&= \left| \sum_a R(a)\mathbb{P}(a|p^*) - \sum_a R(a)\mathbb{P}_{cau}(a|p^*) \right| \\
&= \left| \sum_a R(a)\left[\mathbb{P}(a|p^*) - \mathbb{P}_{cau}(a|p^*)\right] \right|
\end{aligned}
\tag{7}
$$

Then due to the triangle inequality and $0 \le R(a) \le 1$, we have,

$$
\begin{aligned}
&|\mathcal{R}_{\mathbb{P}}(p^*) - \mathcal{R}_{\mathbb{P}_{cau}}(p^*)| \le \sum_a |\mathbb{P}(a|p^*) - \mathbb{P}_{cau}(a|p^*)| \\
&= \sum_a \left| \frac{\lambda\mathbb{P}_{cau}([p^*,a]) + (1-\lambda)\mathbb{P}_{end}([p^*,a])}{\lambda\mathbb{P}_{cau}(p^*) + (1-\lambda)\mathbb{P}_{end}(p^*)} - \frac{\mathbb{P}_{cau}([p^*,a])}{\mathbb{P}_{cau}(p^*)} \right| \\
&= \sum_a \left| \frac{[\lambda\mathbb{P}_{cau}([p^*,a]) + (1-\lambda)\mathbb{P}_{end}([p^*,a])]\mathbb{P}_{cau}(p^*) - [\lambda\mathbb{P}_{cau}(p^*) + (1-\lambda)\mathbb{P}_{end}(p^*)]\mathbb{P}_{cau}([p^*,a])}{[\lambda\mathbb{P}_{cau}(p^*) + (1-\lambda)\mathbb{P}_{end}(p^*)]\mathbb{P}_{cau}(p^*)} \right| \\
&= \sum_a \left| \frac{(1-\lambda)\mathbb{P}_{end}([p^*,a])\mathbb{P}_{cau}(p^*) - (1-\lambda)\mathbb{P}_{end}(p^*)\mathbb{P}_{cau}([p^*,a])}{[\lambda\mathbb{P}_{cau}(p^*) + (1-\lambda)\mathbb{P}_{end}(p^*)]\mathbb{P}_{cau}(p^*)} \right| \\
&= \sum_a \frac{\mathbb{P}_{end}(p^*)}{\mathbb{P}_{cau}(p^*)} \cdot (1-\lambda) \cdot \left| \frac{\frac{\mathbb{P}_{end}([p^*,a])}{\mathbb{P}_{end}(p^*)}\mathbb{P}_{cau}(p^*) - \mathbb{P}_{cau}([p^*,a])}{\lambda\mathbb{P}_{cau}(p^*) + (1-\lambda)\mathbb{P}_{end}(p^*)} \right|
\end{aligned}
\tag{8}
$$

Due to the triangle inequality and $0 < \lambda < 1$, we have

$$
\begin{aligned}
|\mathcal{R}_{\mathbb{P}}(p^*) - \mathcal{R}_{\mathbb{P}_{cau}}(p^*)| &\le \sum_a \frac{\mathbb{P}_{end}(p^*)}{\mathbb{P}_{cau}(p^*)} \cdot (1-\lambda) \cdot \left\{ \frac{\left|\frac{\mathbb{P}_{end}([p^*,a])}{\mathbb{P}_{end}(p^*)}\mathbb{P}_{cau}(p^*)\right| + |\mathbb{P}_{cau}([p^*,a])|}{\lambda\mathbb{P}_{cau}(p^*)} \right\} \\
&= \frac{1-\lambda}{\lambda}\frac{\mathbb{P}_{end}(p^*)}{\mathbb{P}_{cau}(p^*)} \cdot \sum_a \left\{ \mathbb{P}_{end}(a|p^*) + \mathbb{P}_{cau}(a|p^*) \right\} \\
&= \frac{2(1-\lambda)}{\lambda}\frac{\mathbb{P}_{end}(p^*)}{\mathbb{P}_{cau}(p^*)}.
\end{aligned}
\tag{9}
$$

Then we need to prove the upper bound for $\frac{\mathbb{P}_{end}(p^*)}{\mathbb{P}_{cau}(p^*)}$,

$$
\begin{aligned}
&\frac{\mathbb{P}_{end}(p^*)}{\mathbb{P}_{cau}(p^*)} \\
&= \frac{\mathbb{P}_{end}([q^{(1)}, a^{(1)}, \cdots, q^{(i-1)}, a^{(i-1)}, q])}{\mathbb{P}_{cau}([q^{(1)}, a^{(1)}, \cdots, q^{(i-1)}, a^{(i-1)}, q])} \\
&= \frac{\mathbb{P}_{end}(q|[q^{(1)}, a^{(1)}, \cdots, q^{(i-1)}, a^{(i-1)}])}{\mathbb{P}_{cau}(q|[q^{(1)}, a^{(1)}, \cdots, q^{(i-1)}, a^{(i-1)}])} \cdot \frac{\mathbb{P}_{end}([q^{(1)}, a^{(1)}, \cdots, q^{(i-1)}, a^{(i-1)}])}{\mathbb{P}_{cau}([q^{(1)}, a^{(1)}, \cdots, q^{(i-1)}, a^{(i-1)}])}
\end{aligned}
\tag{10}
$$

With the assumption stated in Eqn (3), we have

$$
\begin{aligned}
\frac{\mathbb{P}_{end}(p^*)}{\mathbb{P}_{cau}(p^*)} &= \frac{\mathbb{P}_{end}([q^{(1)}, a^{(1)}, \cdots, q^{(i-1)}, a^{(i-1)}])}{\mathbb{P}_{cau}([q^{(1)}, a^{(1)}, \cdots, q^{(i-1)}, a^{(i-1)}])} \\
&\frac{\mathbb{P}_{end}(a^{(i-1)}|[q^{(1)}, a^{(1)}, \cdots, q^{(i-1)}])}{\mathbb{P}_{cau}(a^{(i-1)}|[q^{(1)}, a^{(1)}, \cdots, q^{(i-1)}])} \cdot \frac{\mathbb{P}_{end}([q^{(1)}, a^{(1)}, \cdots, q^{(i-1)}])}{\mathbb{P}_{cau}([q^{(1)}, a^{(1)}, \cdots, q^{(i-1)}])}
\end{aligned}
\tag{11}
$$

Due to the robust assumption in Eqn (4), we have

$$\frac{\mathbb{P}_{end}(p^*)}{\mathbb{P}_{cau}(p^*)} = \frac{\mathbb{P}_{end}(a^{(i-1)}|q^{(i-1)})}{\mathbb{P}_{cau}(a^{(i-1)}|q^{(i-1)})} \cdot \frac{\mathbb{P}_{end}([q^{(1)}, a^{(1)}, \cdots, q^{(i-1)}])}{\mathbb{P}_{cau}([q^{(1)}, a^{(1)}, \cdots, q^{(i-1)}])}$$

$$= \frac{\mathbb{P}_{end}(a^{(i-1)}|q^{(i-1)})}{\mathbb{P}_{cau}(a^{(i-1)}|q^{(i-1)})} \cdot \frac{\mathbb{P}_{end}(q^{(i-1)}|[q^{(1)}, a^{(1)}, \cdots, q^{(i-2)}])}{\mathbb{P}_{cau}(q^{(i-1)}|[q^{(1)}, a^{(1)}, \cdots, q^{(i-2)}])} \cdot \frac{\mathbb{P}_{end}([q^{(1)}, a^{(1)}, \cdots, q^{(i-2)}])}{\mathbb{P}_{cau}([q^{(1)}, a^{(1)}, \cdots, q^{(i-2)}])} \quad (12)$$

$$= \frac{\mathbb{P}_{end}(a^{(i-1)}|q^{(i-1)})}{\mathbb{P}_{cau}(a^{(i-1)}|q^{(i-1)})} \cdot \frac{\mathbb{P}_{end}([q^{(1)}, a^{(1)}, \cdots, q^{(i-2)}])}{\mathbb{P}_{cau}([q^{(1)}, a^{(1)}, \cdots, q^{(i-2)}])}$$

Iteratively do the above operation, and we have

$$\frac{\mathbb{P}_{end}(p^*)}{\mathbb{P}_{cau}(p^*)} = \prod_{j=1}^{i-1} \frac{\mathbb{P}_{end}(a^{(j)}|q^{(j)})}{\mathbb{P}_{cau}(a^{(j)}|q^{(j)})}$$

$$\leq \prod_{i=1}^{k} e^{-\Delta} \quad (13)$$

$$= e^{-k\Delta}.$$

From Eqn (4), we have

$$\mathcal{R}_{\mathbb{P}_{cau}}([S^{(<i)}, q]) = \sum_{a} R(a)\mathbb{P}_{cau}(a|[S^{(<i)}, q]) = \sum_{a} R(a)\mathbb{P}_{cau}(a|q) = \mathcal{R}_{\mathbb{P}_{cau}}(q). \quad (14)$$

Therefore, following Eqn (9), we have

$$\mathcal{R}_{\mathbb{P}}([S^{(<i)}, q]) = \mathcal{R}_{\mathbb{P}}(q) \geq \mathcal{R}_{\mathbb{P}_{cau}}([S^{(<i)}, q]) - \frac{2(1-\lambda)}{\lambda} \cdot \frac{\mathbb{P}_{end}(p^*)}{\mathbb{P}_{cau}(p^*)} \geq \mathcal{R}_{\mathbb{P}_{cau}}(q) - \frac{2(1-\lambda)}{\lambda} \cdot e^{-k\Delta}.$$
$$(15)$$

For $k \geq \frac{1}{\Delta} \log \frac{2(1-\lambda)}{\epsilon\lambda}$, we have

$$\mathcal{R}_{\mathbb{P}}([S^{(<i)}, q]) \geq \mathcal{R}_{\mathbb{P}_{cau}}(q) - \frac{2(1-\lambda)}{\lambda}\left(\frac{\epsilon\lambda}{2(1-\lambda)}\right) = \mathcal{R}_{\mathbb{P}_{cau}}(q) - \epsilon. \quad (16)$$

$$\square$$

