# OpenReview forum: "SelfCAD: Protecting Your Efficient Reasoning Capabilities via Self Cautious Insertion"
_ICLR.cc/2026/Conference — Submitted to ICLR 2026_

### Official Review · Reviewer_x2de · 2025-10-28

**Soundness:** 3
**Presentation:** 2
**Contribution:** 2
**Rating:** 6
**Confidence:** 4

**Summary:**

The paper introduces SelfCAD (Self-Cautious Anti-Distillation), a method to protect large reasoning models (LRMs) from unauthorized distillation. The key insight is that self-cautious sentences in reasoning traces significantly affect both efficiency and accuracy of distilled models. The authors propose inserting additional self-cautious sentences after generation to make distilled models inefficient (producing 1.2-4.8× longer outputs) while preserving semantic clarity for legitimate users.

**Strengths:**

1. Novel perspective: Identifying self-cautious sentences as a key factor in reasoning efficiency is an interesting observation
2. Lightweight implementation: Inference-time processing without model modification is practical
3.Comprehensive experiments: Testing across multiple models and datasets shows consistency
4. Theoretical grounding: Provides mathematical analysis explaining the mechanism
5. Timely problem: Addresses important concerns about LLM intellectual property

**Weaknesses:**

1. No adaptive attack evaluation: Doesn't test against adversaries who might detect the pattern
2. Strong assumptions: Theoretical analysis assumes distributions remain stable (Eq. 4) without justification
3. Presentation issues: Poor writing quality, structural problems, missing implementation details

**Questions:**

1. Robustness to preprocessing: How does SelfCAD perform when adversaries use simple regex or pattern matching to remove the inserted sentences before distillation?

2. Template variations: Have you tested different self-cautious templates or randomized insertions to make detection harder?

3. Adaptive attacks: Can you evaluate against adversaries who train classifiers to detect artificially inserted vs. natural self-cautious sentences?

---

> ### Author Response · Authors · 2025-11-23
> **Reply to Reviewer x2de**
>
> **W1. No adaptive attack evaluation: Doesn't test against adversaries who might detect the pattern**
>
> Our experiments in Section 4.5 already demonstrate that even GPT4 cannot detect our SelfCAD.
> Besides we also conduct experiments against many adversaries:
>
> 1. Filtering all self-cautious
> Filtering methods would remove the original cautious sentences. As demonstrated in our experiments in Section 3.2, an excessive number of cautious sentences leads to overly lengthy responses. However, removing all cautious sentences would also result in a performance drop, which is also a kind of protection. Therefore, simply paraphrasing or filtering methods cannot lead the distillation work better again.
> The method of truncating intermediate steps is similar to the filtering method. As claimed in Section 3.2, removing any part of the Chain of Thought (CoT) leads to a decline in performance.
> 2. Inference time “Efficient Reasoning methods.”
>
>     We also conduct experiments with the DEER[A], an inference trancated method which claimed to be effective on removing the unnecessary thoughts. The hyper parameter can chosen as their default. However, as the results show, they are not useful for our AntiDistill.
>
>     | Options | Original Inference |
>     | --- | --- |
>     | Vanilla | 600 |
>     | AntiDistill | 2900 |
>     | AntiDistill+DEER | 2800 |
>
> **Reference**
>
> [A] Dynamic Early Exit in Reasoning Models
>
> **W2. Strong assumptions: Theoretical analysis assumes distributions remain stable (Eq. 4) without justification.**
> The stability is also practical as LLM’s generation distribution are formed during training, their pattern are stably found in many works[A] called “Natural Fingerprint”.  Therefore, LLMs will not change their generation distribution for self-cautious.
>
> **Reference**
>
> [A] Natural Fingerprints of Large Language Models
>
> **W3. Presentation issues: Poor writing quality, structural problems, missing implementation details**
> We revise our paper by removing some unnecessary bold sentences. As for the implementation details, we report the algorithm in Alogrithm 1,  dataset, model, inserted sentence, training settings(learning rate, training epoch) in Section 4.1 and the prompt for GPT evaluation in the appendix. We believe it is enough for implementation as our method do not have many hyper-parameters. We welcome your feedback on any missing parameters and will respond promptly.
>
> **Q1. Robustness to preprocessing: How does SelfCAD perform when adversaries use simple regex or pattern matching to remove the inserted sentences before distillation?**
>
> Pattern matching methods would remove the original cautious sentences. As demonstrated in our experiments in Section 3.2, an excessive number of cautious sentences leads to overly lengthy responses. However, removing all cautious sentences would also result in a performance drop. Therefore, we believe that simple regex or pattern matching methods cannot be used to bypass our defense.
>
> **Q2. Template variations: Have you tested different self-cautious templates or randomized insertions to make detection harder?**
>
> Sure, we’ve test randomised choose 11 protected self-cautious sentences to make the detection harder. The results are shown as follows, the results show that adding different self-cautious templates can also make the LLM generate longer, but the overall performance is lower than constant insertion.
>
> | Options | Folio | PiQA | ProofWriter | Social_iQA | GSM8K | Average Improvement |
> | --- | --- | --- | --- | --- | --- | --- |
> | Vanilla | 900 | 500 | 1000 | 450 | 600 | - |
> | AntiDistill | 3300(3.7x) | 1300(2.6x) | 3000(3x) | 1300(2.9x) | 2900(4.8) | 3.4x |
> | AntiDistill (Random Insert Sentence) | 1400(1.6x) | 2400(4.8x) | 2100(2.1x) | 2000(4.4x) | 1800(3x) | 3.2x |
>
> **Q3. Adaptive attacks: Can you evaluate against adversaries who train classifiers to detect artificially inserted vs. natural self-cautious sentences?**
>
> Training a classifier is nearly infeasible, as it requires an extremely large amount of manually annotated data. Moreover, different models have distinct generation habits, making it difficult to achieve generalizability.
>
> Additionally, the experiment in Section 4.5 can be considered a form of prompt-based detection, and even state-of-the-art models struggle to distinguish between artificially inserted and natural self-cautious sentences.

---

### Official Review · Reviewer_NMMS · 2025-10-28

**Soundness:** 2
**Presentation:** 3
**Contribution:** 2
**Rating:** 4
**Confidence:** 4

**Summary:**

This paper proposes SelfCAD, a defense mechanism against unauthorized reasoning distillation. The key idea is to inject self-cautious sentences into chain-of-thought explanations—phrases expressing doubt or self-verification. These interventions increase the verbosity and reduce the efficiency of student models trained on the generated traces, while preserving the original model’s transparency and performance. Experiments on mathematical reasoning show that distillation from SelfCAD-protected data leads to significantly longer reasoning chains and modestly reduced accuracy.

**Strengths:**

1. Simple and low-cost deployment — does not require modifying the trainer or model architecture.

2. Maintains transparency — unlike encryption or CoT suppression, users still see the reasoning process.

3. Applies to real-world API scenarios — defense could plausibly be adopted by service providers immediately.

**Weaknesses:**

The most important thing is that I think defense seems trivial to bypass. While the idea is intuitive, the core mechanism—adding self-cautious phrases into reasoning traces—appears easily removable. A defender with basic text-processing capability (e.g., filtering, paraphrasing, style normalization) could simply strip or rewrite these caution sentences before distillation. This raises a fundamental question:

If the protection can be removed by a simple text post-processing step, is the defense truly effective?

Additionally, because the defense operates purely at the surface-form level, an attacker could bypass it entirely by:

• distilling from logits / token probabilities instead of CoT text

• accessing hidden states directly (a common research practice)

• supervised fine-tuning student confidence calibration back to normal

The paper would benefit from a more rigorous robustness evaluation, including adversarial distillation settings rather than only naïve student training pipelines. So far the method appears fragile under even basic threat models.

**Questions:**

See weakness.

---

> ### Author Response · Authors · 2025-11-23
> **Reply to Reviewer NMMS**
>
> Thank you for raising these important concerns. However, we believe that simple text-processing capabilities are insufficient to bypass our defense. Merely filtering out self-cautious sentences would lead to a decline in model performance. As we emphasized in Section 3.2, while an excessive number of self-cautious sentences may result in unnecessarily lengthy responses, an appropriate amount of cautious language is beneficial. The self-cautious sentences we introduce are intentionally blended with the beneficial cautious expressions naturally generated by the teacher model, making them inseparable.
>
> Regarding your concern that our defense operates only at the surface-form level, we believe there may be a misunderstanding of our experimental setting. Our goal is to protect the intellectual property of API providers by preventing the misuse of their generated responses. The protection of open-source models falls outside the scope of our work. Therefore, the three attack methods you mentioned—accessing hidden states directly, supervised fine-tuning to recalibrate student confidence and distilling from logits or token probabilities—require a white-box setting, which is not applicable in our scenario. Moreover, the suggestion of "distilling from logits/token probabilities instead of CoT text" seems particularly unusual, as training a reasoning model without access to CoT text would make obtaining CoT logits even more infeasible.
>
> However, we also conduct experiments with a method related to your third option, called DEER[A], an inference method which shortens LLMs’ output using their confidence. The hyper parameter can chosen as their default. However, as the results show, they are not useful for our AntiDistill, demonstrating the robustness of our method.
>
> | Options | Original Inference |
> | --- | --- |
> | Original | 600 |
> | SelfCAD | 2900 |
> | SelfCAD+DEER | 2800 |
>
> **Reference**
>
> [A] Dynamic Early Exit in Reasoning Models

---

### Official Review · Reviewer_KNkw · 2025-11-01

**Soundness:** 2
**Presentation:** 3
**Contribution:** 2
**Rating:** 4
**Confidence:** 3

**Summary:**

This paper proposes **SelfCAD (Self-Cautious Anti-Distillation)**, a lightweight defense mechanism designed to protect reasoning-capable LLMs from unauthorized distillation while preserving transparency.
The authors first perform a fine-grained analysis of reasoning trajectories, decomposing them into *statement*, *reasoning*, *self-cautious*, and *conclusion* sentences. They discover that self-cautious sentences—phrases like “wait” or “let me double-check”—strongly influence both the *efficiency* (output length) and *accuracy* of reasoning models.
Leveraging this insight, SelfCAD strategically inserts additional self-cautious sentences into reasoning traces. This manipulation keeps human readability intact but leads student models trained on these traces to produce **redundant and inefficient reasoning**, thereby degrading the effectiveness of model distillation.
Experiments across Llama-3.2-1B/3B and Qwen2.5-1.5B/7B show that distillation with SelfCAD-processed traces increases output length by 1.3–4.8× and reduces accuracy by 2–8% while maintaining over 99% semantic equivalence for humans.
The paper offers a practical and creative perspective on proactive anti-distillation for reasoning models.

**Strengths:**

- **Conceptual novelty:** Introduces a unique efficiency-oriented defense mechanism distinct from watermarking or audit-based approaches.
- **Lightweight and practical:** Can be applied at inference time without model retraining or architecture modification.
- **Empirical clarity:** Includes ablations and trajectory analyses that clearly illustrate the “self-cautious effect.”
- **Transparency preserved:** Maintains semantic fidelity while effectively degrading distillation efficiency.
- **Reproducibility:** Implementation details and evaluation settings are well documented.

**Weaknesses:**

- **Shallow methodological contribution:** The proposed method essentially inserts fixed *self-cautious* sentences (e.g., “*Wait, let me check again…*”) after each reasoning step. There is no adaptive component, learning mechanism, or optimization objective. As a result, the core technique is heuristic and lacks algorithmic or theoretical depth.
- **Overstated novelty:** While the problem of protecting reasoning traces from distillation is timely and relevant, the proposed solution is minimal and does not introduce new principles beyond simple text augmentation. Most of the originality lies in the *problem framing* rather than the *technical approach*.
- **Lack of rigorous baselines:** The paper does not compare SelfCAD against simpler or more intuitive baselines—such as random phrase insertion, noise injection, or partial reasoning truncation—that could yield similar degradation effects.
- **Limited generalization:** All experiments focus solely on mathematical reasoning tasks. It remains unclear whether the proposed mechanism would transfer to other domains such as commonsense reasoning, logical inference, or code generation.
- **Weak theoretical justification:** The included theorem is more descriptive than analytical—it essentially formalizes an intuitive observation that adding self-cautious sentences encourages longer reasoning. No provable guarantees or quantitative bounds are provided.
- **Formatting and polish issues:** Several sections are overly bolded or inconsistently formatted, which negatively impacts readability. These should be fixed for the camera-ready version.

**Questions:**

1. Since the core mechanism is inserting templated *“wait”* phrases, how does SelfCAD differ from random or semantically neutral text insertion? Could a trivial baseline achieve similar anti-distillation effects?
2. Have you evaluated whether paraphrasing or filtering the outputs before student training can remove these self-cautious tokens and thus bypass the defense?
3. Does the anti-distillation effect persist when the student model is fine-tuned for more epochs or trained with RLHF-based objectives instead of simple supervised distillation?
4. Beyond math reasoning, do the authors have evidence that SelfCAD generalizes to domains such as commonsense QA, coding, or multimodal reasoning?
5. Could a future version of SelfCAD learn *where* and *when* to insert self-cautious sentences adaptively, rather than applying them uniformly across all reasoning steps?

---

> ### Author Response · Authors · 2025-11-23
> **Reply to Reviewer KNkw (1/2)**
>
> **W1 & W2. Shallow methodological contribution & Overstated novelty**
>
> Thank you for your concerns regarding the contribution and novelty of our method. Our approach does not involve an optimization objective or similar components, which also signifies its advantages in preserving the performance of the teacher model and being straightforward and practical. As demonstrated in Tables 3, 4, and 5, our method achieves excellent results with only minimal processing of the output text. Moreover, we have provided comprehensive theoretical proofs and analyses to support it. In terms of novelty, we are the first work to utilize the reasoning model's chain-of-thought length for defense and the first to conduct a detailed analysis of it. In fact, the ease of deployment is an advantage of our approach, as it can be quickly applied to any LLM API without incurring additional computational overhead.  Therefore, we believe our approach will bring sufficient benefits and insight to the entire community.
>
> **W3 & Q1. Baseline**
>
> Thank you for your concerns and suggestions regarding the baseline. We have added a baseline using random noise by inserting random, semantically neutral phrases like "Fine" or "Got it" in the middle of sentences. We fine-tuned Llama3.2-1B-Instruct and evaluated it on the GSM8K dataset. The experimental results are as follows:
>
> | Llama3.2-1B | Unprotected | Random Noise | SelfCAD |
> | --- | --- | --- | --- |
> | Response Length | 600 | 800 | 1300 |
>
> We observed that although inserting random noise slightly increased the response length, the performance was significantly inferior to our SelfCAD method.
>
> Apart from the baseline method, we also conduct experiments on Qwen2.5-1.5B with the DEER[A], an inference trancated method which claimed to be effective on removing the unnecessary thoughts. However, as the results show, they are not useful for our SelfCAD method.
>
> | Options | Response Length |
> | --- | --- |
> | Original | 600 |
> | SelfCAD | 2900 |
> | SelfCAD+DEER | 2800 |
>
> **W4. Weak theoretical justification: The included theorem is more descriptive than analytical—it essentially formalizes an intuitive observation that adding self-cautious sentences encourages longer reasoning. No provable guarantees or quantitative bounds are provided.**
>
> Our theoretical analysis is not descriptive. First, we need to clarify that our theorem doesn’t mean all self-cautious lead to longer generation. Our theorem analyse the common circumstance that correct reasoning step being over-cautious, and we find that if such a scenario happens
> over a threshold, the model will tend to over-cautious on latter correct reasoning steps and lead to overthinking. Such a result also guarantees that our SelfCAD can make LLMs overthink and prevent unauthorized overthinking.
>
> **W5. Formatting and polish issues: Several sections are overly bolded or inconsistently formatted, which negatively impacts readability. These should be fixed for the camera-ready version.**
>
> Thank you for the reminder. We will address this issue in future updates.
>
> **Q2. Have you evaluated whether paraphrasing or filtering the outputs before student training can remove these self-cautious tokens and thus bypass the defense?**
>
> Filtering methods would remove the original cautious sentences. As demonstrated in our experiments in Section 3.2, an excessive number of cautious sentences leads to overly lengthy responses. However, removing all cautious sentences would also result in a performance drop. Therefore, we believe that simple paraphrasing or filtering methods cannot be used to bypass our defense.
>
> **Q3. Does the anti-distillation effect persist when the student model is fine-tuned for more epochs or trained with RLHF-based objectives instead of simple supervised distillation?**
>
> We list the output length for GSM8K with Qwen after training with different epochs. From the results, one can see that our method can consistently increase model’s output length.
>
> | Options | Original Inference(3epoch) | SelfCAD(1epoch) | SelfCAD(2epoch) | SelfCAD(3epoch) | SelfCAD(4epoch) |
> | --- | --- | --- | --- | --- | --- |
> | Response Length | 600 | 2100 | 2600 | 2900 | 3200 |
>
> As for the question about training with RLHF-based objectives, we believe that current distillation methods have not yet incorporated RLHF-based objectives, or such approaches have not been widely recognized in the academic community.

---

> ### Author Response · Authors · 2025-11-23
> **Reply to Reviewer KNkw (2/2)**
>
> **Q4. Beyond math reasoning, do the authors have evidence that SelfCAD generalizes to domains such as commonsense QA, coding, or multimodal reasoning?**
>
> Sure, we conduct further evaluations on Folio, piqa, proof writer, and socialqa dataset with Qwen. The generation length are shown as follow. From the results, one can see that our SelfCAD can consistently make the distilled model ineffective.
>
> | Options | Folio | PiQA | ProofWriter | Social_iQA |
> | --- | --- | --- | --- | --- |
> | Original | 900 | 500 | 1000 | 450 |
> | SelfCAD | 3300(3.7x) | 1300(2.6x) | 3000(3x) | 1300(2.9x) |
>
> **Q5. Could a future version of SelfCAD learn *where* and *when* to insert self-cautious sentences adaptively, rather than applying them uniformly across all reasoning steps?**
>
> Thank you for your suggestion regarding the adaptive aspect of our method. However, our goal is to provide a solution that can be deployed to any LLM API without introducing additional overhead. As described in the paper, our method only requires simple processing on the CPU to efficiently protect the response. Our experimental results demonstrate the efficiency of our method, while the findings in Section 4.5 also confirm its stealthiness.

---

### Official Review · Reviewer_qVvB · 2025-11-01

**Soundness:** 2
**Presentation:** 1
**Contribution:** 2
**Rating:** 2
**Confidence:** 3

**Summary:**

This paper introduces SelfCAD, a lightweight, inference-time defense that preserves transparent reasoning traces while degrading the effectiveness of unauthorized distillation. The key insight is that the number of self-cautious sentences in reasoning trajectories critically impacts both efficiency and accuracy in distilled models. SelfCAD post-processes teacher outputs by inserting self-cautious sentences after each reasoning step, keeping the original reasoning intact for human and LLM auditors but causing student models trained on these traces to become less confident and over-verbose. Analyses show self-cautious sentences strongly reduce sequence termination probability and lengthen trajectories; training-time studies confirm that removing self-cautious sentences shortens outputs while adding them increases length and can reduce accuracy. A simple theoretical model explains how repeated self-cautiousness induces excessive reasoning.

**Strengths:**

(1) The paper identifies and isolates the role of self-cautious sentences in driving reasoning length and student confidence, providing both empirical and theoretical support for an efficiency-oriented protection mechanism.

(2) SelfCAD is practical and minimally invasive: it operates as post-processing at inference time, requires no teacher fine-tuning, preserves original reasoning content, and can be run on CPU in minutes.

(3) The methodology is clearly described, including sentence-type categorization, termination-probability analysis, a simple but principled theorem, and an explicit algorithm for insertion.

(4) Experiments span multiple student sizes and two distillation sources, with consistent increases in inference cost and modest accuracy reductions; the stealthiness evaluation suggests preserved semantic transparency for users and auditors.

**Weaknesses:**

(1) The insertion strategy is uniform and naive, applying the same self-cautious sentence after every step; more targeted placement tuned to correctness or step importance could strengthen effect or reduce accuracy loss, but is not explored.

(2) The defense primarily targets text-reasoning math datasets; generalization to other domains and modalities, longer-context tasks, or instruction-heavy settings is not demonstrated.

(3) Adversary adaptivity is not studied: a distiller could filter or downweight self-cautious spans, apply truncation, use RLHF to penalize over-caution, or use contrastive objectives to recover efficiency.

(4) The semantic-equivalence “stealth” check uses LLM judges on a binary yes/no criterion; human studies or more granular equivalence metrics would better validate transparency preservation.

(5) Theoretical assumptions (mixture model, stability to prior steps) simplify dynamics; empirical tests that vary $\lambda$ and distributional separability would strengthen the causal link to observed length inflation.

(6) Potential collateral effects on downstream evaluators or audit tools that rely on reasoning brevity or structure are not assessed.

**Questions:**

(1) How robust is SelfCAD to adaptive distillers that strip or downweight self-cautious sentences, truncate intermediate steps, or apply RLHF to penalize excessive caution; can you report results under such countermeasures?

(2) Can you design and evaluate a selective insertion policy that targets steps likely to be correct or pivotal, using lightweight heuristics or a small classifier, to maximize length inflation while minimizing accuracy degradation?

(3) How does SelfCAD perform beyond math, for example, in symbolic logic, code reasoning, tool-use chains, or long-context QA, and does the effect size persist with longer contexts and different tokenizers?

(4) Could you report human evaluation of transparency and usefulness, beyond LLM-judge equivalence, including perceived clarity, redundancy, and auditability of the modified traces?

(5) What are the impacts on student training stability and compute cost during distillation, such as gradient variance, convergence speed, and GPU hours; can you quantify the additional cost imposed on the distiller?

(6) Can you provide ablations on the content and style of self-cautious sentences, frequency of insertion, and position relative to sub-steps, to identify minimally invasive yet maximally effective variants?

---

> ### Author Response · Authors · 2025-11-23
> **Reply to Reviewer qVvB (1/2)**
>
> **W1 & Q2. The insertion strategy is uniform and naive, applying the same self-cautious sentence after every step; more targeted placement tuned to correctness or step importance could strengthen effect or reduce accuracy loss, but is not explored. Can you design and evaluate a selective insertion policy that targets steps likely to be correct or pivotal, using lightweight heuristics or a small classifier, to maximize length inflation while minimizing accuracy degradation?**
>
> Thank you for your suggestion regarding the adaptive aspect of our method. However, our goal is to provide a solution that can be deployed to any LLM API without introducing additional overhead. As described in the paper, our method only requires simple processing on the CPU to efficiently protect the response. Our experimental results demonstrate the efficiency of our method, while the findings in Section 4.5 also confirm its stealthiness.
>
> Additionally, regarding your mention of "maximize length inflation while minimizing accuracy degradation," we believe this is also a misunderstanding of our article's purpose. Our primary goal is to maximize length inflation, while accuracy degradation is also something we view favorably, though it was not the main focus of our algorithm.
>
> **W2 & Q2. SelfCAD perform beyond math**
>
> Sure, we conduct further evaluations on Folio, piqa, proof writer, and socialqa dataset with Qwen. The generation length are shown as follow. From the results, one can see that our SelfCAD can consistently make the distilled model ineffective.
>
> | Options | Folio | PiQA | ProofWriter | Social_iQA |
> | --- | --- | --- | --- | --- |
> | Original | 900 | 500 | 1000 | 450 |
> | SelfCAD | 3300(3.7x) | 1300(2.6x) | 3000(3x) | 1300(2.9x) |
>
> **W3 & Q1. Adversary adaptivity:**
>
> Filtering methods would remove the original cautious sentences. As demonstrated in our experiments in Section 3.2, an excessive number of cautious sentences leads to overly lengthy responses. However, removing all cautious sentences would also result in a performance drop, which is also a kind of protection. Therefore, simply paraphrasing or filtering methods cannot lead the distillation work better again.
>
> As for truncated method, we also conduct experiments with the DEER[A], an inference trancated method which claimed to be effective on removing the unnecessary thoughts. The hyper parameter can chosen as their default. However, as the results show, they are not useful for our SelfCAD method.
>
> | Options | Response Length |
> | --- | --- |
> | Original | 600 |
> | SelfCAD | 2900 |
> | SelfCAD+DEER | 2800 |
>
> Regarding the latter two methods you mentioned—using RLHF to penalize over-caution or employing contrastive objectives to recover efficiency—no similar approaches have emerged in academia thus far. If there are any related works, please point them out.
>
> **Reference**
>
> [A] Dynamic Early Exit in Reasoning Models
>
> **W4 & Q4. Human evaluation of transparency and usefulness**
>
> We selected 100 data points and had three researchers annotate them, with the final result determined by majority vote. The labeling was designed as a single-choice task with three options: A. Affects readability, B. Does not affect readability but differences are noticeable, C. Almost no difference. The annotation results are as follows:
>
> | Options | A | B | C |
> | --- | --- | --- | --- |
> | Rate | 0 | 21 | 79 |
>
> **W5. Theoretical assumptions (mixture model, stability to prior steps) simplify dynamics; empirical tests that vary $\lambda$ and distributional separability would strengthen the causal link to observed length inflation:**
>
> As for the assumptions, we believe the mixture model is reasonable as the reasoning traces only contain the two types (ending or self-cautious) after the reasoning step as we former defined. And the stability is also practical as LLM’s generation distribution are formed during training, their pattern are stably found in many works[A] called “Natural Fingerprint”.
>
> As for the empirical evaluation, we note that $\lambda$ is formed in models’ pre-training and post-training. Due to the resource limit, we cannot retrain the model to control $\lambda$ or even estimate the exact number of it. However, we believe it does not affect the contribution of our claims, as our theoretical analysis proved that no matter model’s $\lambda$ is set to which, generating too many self-cautious sentences will finally lead to answer.
>
> And the separability cannot be controlled as the ending sentences and self-cautious are totally different as shown in our Table 1. Sentence “Wait, let’s check” will not shown in the ending generation. Therefore, they are totally separable and cannot be empirically controlled.
>
> **Reference**
>
> [A] Natural Fingerprints of Large Language Models

---

> ### Author Response · Authors · 2025-11-23
> **Reply to Reviewer qVvB (2/2)**
>
> **W6. Potential collateral effects on downstream evaluators or audit tools that rely on reasoning brevity or structure are not assessed:**
>
> First, our experiments in 4.5 demonstrates that such generation cannot be detected by SoAT commercial LLMs like GPT. Therefore,  auditing cannot make our SelfCAD fail.
> Secondly, our former experiments in **W3 & Q1** also demonstrate that our SelfCAD’s robustness against adversarial strategies at the training or inference time.
>
> **Q5. What are the impacts on student training stability and compute cost during distillation, such as gradient variance, convergence speed, and GPU hours; can you quantify the additional cost imposed on the distiller?**
>
> The training cost for SelfCAD and vanilla distillation is almost the same. We train Qwen on Oss distilled dataset with $4$ A100 40G. The average time cost for each epoch is as follows:
>
> | Original | 1.33h |
> | --- | --- |
> | SelfCAD | 1.33h |
>
> The reason is we just add some sentences in the corpus, the additional length of training tokens are negligible compared with the training cost. As for the training loss and gradient norm, we list them as follows,
>
> | Epochs | 0.01 | 0.5 | 1.0 | 1.5 | 2.0 | 2.5 | 3.0 |
> | --- | --- | --- | --- | --- | --- | --- | --- |
> | Original | 1.27 | 0.72 | 0.76 | 0.62 | 0.6 | 0.5 | 0.51 |
> | SelfCAD | 1.31 | 0.74 | 0.71 | 0.57 | 0.58 | 0.53 | 0.51 |
>
> | Epochs | 0 | 0.5 | 1.0 | 1.5 | 2.0 | 2.5 | 3.0 |
> | --- | --- | --- | --- | --- | --- | --- | --- |
> | Original | 2.17 | 1.15 | 1.06 | 1.08 | 1.08 | 1.15 | 1.15 |
> | SelfCAD | 2.23 | 0.97 | 1.02 | 0.99 | 0.97 | 1.09 | 1.07 |
>
> From the results, one can see that the training process of SelfCAD is similar to Original distillation. The model will stably converge to the final.
>
> **Q6. Can you provide ablations on the content and style of self-cautious sentences, frequency of insertion, and position relative to sub-steps, to identify minimally invasive yet maximally effective variants?**
>
> We conducted an ablation study on the frequency of insertion. Using the Llama3.2-1B-Instruct model, we performed fine-tuning with insertion probabilities of 0, 0.2, 0.5, 0.7, and 1, respectively. The results are shown in the table below.
>
> | Insert probability | Ori | 0.2 | 0.5 | 0.7 | 1.0 |
> | --- | --- | --- | --- | --- | --- |
> | Response Length | 600 | 1000 | 1100 | 1200 | 1300 |
>
> The results show that our SelfCAD can increase reasoning length significantly even with low injection rate, demonstrating our effectiveness on preventing unauthorised distillation.

---

> > ### Comment · Reviewer_qVvB · 2025-11-27
> >
> > I think the experiments and explanations provided by the author completely addressed my concerns. But I still think the presentation is a bit weak, and the author should present the innovation more clearly in the revised version. Based on the author's rebuttal, I will give a positive score.

---

> ### Author Response · Authors · 2025-11-28
>
> We sincerely appreciate Reviewer qVvB's acknowledgment of our rebuttal and the increased rating to 6. In response to the concerns raised, we have further clarified our contributions and innovations in the abstract and introduction of the revised manuscript. We are grateful for the reviewer's constructive feedback, which has significantly enhanced both the rigor and clarity of our paper.

---

### Meta-Review · Area_Chair_5GoN · 2026-01-06

**Summary:**

The paper proposes SelfCAD, an inference-time, training-free defense against black-box distillation of reasoning traces, by inserting self-cautious sentences into chain-of-thought to make downstream distilled students more verbose and less effective while keeping traces readable for users/auditors. The main concerns in reviews were whether the method is too heuristic/simple, whether it is robust to adaptive attackers (e.g., filtering/paraphrasing), and whether results generalize beyond math; presentation quality was also raised.

**Reviewer Concerns:**

Addressed by rebuttal:
+Generalization beyond math: authors report additional evaluations on non-math reasoning/QA datasets and show length inflation persists.

+Adaptive / bypass concerns (text preprocessing): authors argue filtering/paraphrasing cannot bypass without harming performance and add tests including an “efficient reasoning” truncation method (DEER).

+Baselines: authors add a random/neutral phrase insertion baseline and show it is weaker than SelfCAD.

+Transparency validation: authors add a small human study on readability/noticeability.

+Ablations & persistence: authors provide ablations on insertion probability and show the effect persists across more student training epochs.

Concerns are still outstanding:

-Depth/novelty: even after rebuttal, the core mechanism remains a simple text augmentation heuristic; the paper’s strongest contribution is the problem framing + empirical characterization rather than a technically deep defense.

-Robustness scope: the evaluation remains limited to the stated black-box, text-trace distillation threat model; stronger adaptive attacks (e.g., learned detectors/style normalizers, or alternative distillation pipelines) are only partially addressed.

-Presentation: reviewers still note the need for clearer positioning and cleaner writing in the final version.

**Reviewer Scores:**

qVvB: 2 → 6 (explicitly states rebuttal addressed concerns and would give a positive score).

KNkw: 4 → 4–5 (baseline + non-math results address key asks, but “minimal technique/overstated novelty” likely keeps it borderline).

NMMS: 4 → 4–5 (threat model clarification + DEER experiment help, but skepticism about bypassability may persist).

x2de: 6 → 6 (main requests were adaptive-attack evaluation + presentation; rebuttal adds partial robustness evidence but not a full adaptive suite).

---

### Decision · Program_Chairs · 2026-01-26

Reject